**Metabolism Methods Commentary Series**

# Measuring mitochondrial membrane potential

Oscar Tovar-Ferrero [iD] [1,5], Javier Rubio [iD] [1,5], Antonio Zorzano [iD] [2,3,4], Guillermo Martínez-Corrales [iD] [1✉] & Marc Liesa [iD] [1,2✉]

Mitochondrial function determines the balance between health and disease, making understanding their bioenergetics an increasingly important focus across biomedical research. Among the most relevant and accessible measures of mitochondrial function in intact cells is the mitochondrial membrane potential (ΔΨm), typically determined with commercially available fluorescent dyes. However, these probes are prone to misjudgement, as oversimplified interpretation of their fluorescent signal can cause important mistakes during both data acquisition and conclusions. This metabolism methods commentary describes how the interplay between mitochondrial membrane potential, mitochondrial ATP synthesis and oxygen consumption influences ΔΨm as a readout for mitochondrial respiratory function. We then examine how widely used mitochondrial dyes label mitochondria and what their fluorescence really reflects. Finally, we highlight four hallmark principles for the accurate determination of mitochondrial ΔΨm, which we hope will promote the rigorous assessment and interpretation of mitochondrial membrane potential across research fields.

**Subject Categories** Metabolism; Methods & Resources

See also: Metabolism Methods Commentary Series

## Introduction

Mitochondria are at the center of aerobic life, and changes in their respiratory function play a role in multiple diseases. This led to an explosion in the study of mitochondrial bioenergetics by researchers from different disciplines. One of the most accessible approaches to determine mitochondrial function in intact cells is the measurement of mitochondrial membrane potential (ΔΨm) using commercially available fluorescent dyes. This is an attractive approach, as fluorescent dyes are relatively affordable, they enable high-throughput studies, and most laboratories have access to a flow cytometer and a microscope to measure their fluorescence. However, this attractiveness, combined with these dyes being marketed as mitochondrial-specific dyes or mito-probes, can lead to the overly simplistic interpretation that a more fluorescent signal from the dye is unequivocally showing increased mitochondrial function. In this piece, we provide a simplified summary of the relationship between mitochondrial membrane potential, mitochondrial ATP synthesis and oxygen consumption, which is needed to understand how differences in ΔΨm report on mitochondrial respiratory function. We then review the molecular basis of how the most popular mitochondrial dyes can label mitochondria and what their fluorescence can report on. This review is structured as the four hallmarks of mitochondrial ΔΨm measurements that, in our experience, are commonly missed by researchers not familiar with mitochondrial physiology. We hope that this piece will be useful to all current and future metabolism researchers and help democratize the accurate assessment and interpretation of mitochondrial membrane potential measurements across laboratories of different disciplines.

## The first hallmark: divergent changes in oxidative phosphorylation (OXPHOS) can be associated with the same mitochondrial membrane potential (ΔΨm) shift

### OXPHOS is a balance between the generation and consumption of the ΔΨm across the inner mitochondrial membrane

Oxidative phosphorylation (OXPHOS) refers to the process by which mitochondria consume oxygen to synthesize ATP. OXPHOS is constituted by two autonomous modules: the electron transport chain (ETC), which reduces oxygen to water and generates the ΔΨm, and the ATP synthase, which consumes the ΔΨm (Fig. 1A). Electron transport generates the ΔΨm by driving protons out of the mitochondrial matrix to the intermembrane space (IMS), while ATP synthase consumes the ΔΨm by bringing protons from the IMS back to the matrix (Fig. 1A). Importantly, there are other mitochondrial transport processes and reactions that affect the proton gradient and the ΔΨm, meaning that these other processes different that ATP synthase and the ETC can impact OXPHOS. Mitochondria in which the ΔΨm is mostly consumed by the ATP synthase are named coupled mitochondria (see also Glossary), as the ΔΨm generation by the ETC is coupled to ΔΨm consumption by the ATP synthase. Mitochondria in which the contribution of ATP synthase to ΔΨm consumption is decreased are named leaky or inefficient mitochondria, and mitochondria in which ATP synthase has no contribution to ΔΨm consumption are named uncoupled. Consequently, OXPHOS does not occur in uncoupled mitochondria. But how does the consumption of ΔΨm by ATP synthase

[1]Institut de Biologia Molecular de Barcelona, IBMB, CSIC, C/Baldiri Reixac 10, Barcelona 08028 Catalonia, Spain. [2]CIBERDEM, Instituto de Salud Carlos III, Madrid, Spain. [3]Institute for Research in Biomedicine (IRB Barcelona), The Barcelona Institute of Science and Technology, Baldiri Reixac, 10-12, Barcelona, Catalonia, Spain. [4]Departament de Bioquímica i Biomedicina Molecular, Facultat de Biologia, Universitat de Barcelona, Barcelona, Catalonia, Spain. [5]These authors contributed equally: Oscar Tovar-Ferrero, Javier Rubio. ✉E-mail: gmcbmc@ibmb.csic.es; mlrbmc@ibmb.csic.es
https://doi.org/10.1038/s44318-025-00632-9 | Published online: 17 November 2025

control the ETC and therefore mitochondrial oxygen consumption?

**Glossary**

| | |
|---|---|
| **OXPHOS** | Oxidative phosphorylation. The process by which mitochondrial electron transport and oxygen consumption produces new molecules of ATP from ADP. |
| **ETC** | Mitochondrial electron transport chain. |
| **Coupled mitochondria** | Mitochondria in which most of the $\Delta\Psi$m generated by the ETC is consumed by the ATP synthase, meaning that mitochondrial oxygen consumption will be mostly coupled, or linked, to ATP synthesis. |
| **Uncoupled mitochondria** | Mitochondria in which the $\Delta\Psi$m generated by the ETC is dissipated by other processes different than the ATP synthase, to the point that ATP synthesis has no control over the rates of oxygen consumption. |
| **Leaky/inefficient mitochondria** | Mitochondria in which ATP synthesis still controls and dictates the rates of oxygen consumption, but a highly significant fraction of the $\Delta\Psi$m is consumed by other processes. |
| **Proton motive force ($\Delta$p)** | Potential energy generated by the proton gradient across the inner mitochondrial membrane, which is used to synthesize ATP. The |

| | |
|---|---|
| | difference in charge ($\Delta\Psi$m) generated by the $\Delta$p is about 80% of the potential energy, while the difference in pH only accounts for 20%. |
| **Protonophores** | Chemicals that dissipate the proton gradient across membranes, by acting as proton translocators or by generating pores in the membrane letting protons through. |
| **Nernst equation** | It can be used to calculate the $\Delta\Psi$m in millivolts using the fluorescence intensity (FI) of the Nernstian dyes detected in mitochondria in condition 1 and in condition 2; R: constant; T: temperature in Kelvin. $\Delta\Psi$m = (RT) log (FI condition 1/FI condition 2). |
| **Nernstian behavior** | It refers to the capacity of the fluorescence of a dye to report on $\Delta\Psi$m, as the dye distribution into the mitochondria follows the voltage predicted by the Nernst equation. |
| **Quenching** | A process that reduces or eliminates the fluorescence of a molecule, in this case caused by the high density of the same fluorescent molecule (i.e., one molecule absorbs the light emitted by another molecule). |

The ETC is formed by multiprotein complexes and electron carriers, which accept the electrons from nutrient oxidation and transfer them to oxygen. Oxygen consumption occurs when the electrons delivered from complex III of the ETC are used in complex IV to reduce oxygen into water. The transfer of electrons across complexes I, III and IV drives proton translocation across the inner membrane. If the proton concentration in the IMS is too high, electron transport will be impeded because proton translocation driven by electron transfer will be hindered. This ETC-driven proton translocation generates the proton motive force ($\Delta$p) across the inner membrane. $\Delta$p is the potential energy generated by the difference in proton concentrations, $\Delta$pH, and by the difference in charge, $\Delta\Psi$m. Most of the potential energy contained in the $\Delta$p stems from the difference in charge, not from the difference in the concentration of protons: $\Delta\Psi$m is about 80% of the $\Delta$p (Nicholls and Ferguson 2013). The fact that electron transfer drives the translocation of protons from the matrix to the intermembrane space explains why higher rates of electron transfer and the concomitant increase in oxygen consumption can sometimes be associated with an increase of the $\Delta$p and $\Delta\Psi$m (Nicholls and Ferguson 2013). An example is the hyperpolarization of mitochondria and the simultaneous increase in respiration induced by high glucose in pancreatic beta-cells (Fig. 1A vs. B) (Gerencser et al, 2016).

Connecting ATP synthesis to respiration, the ATP synthase consumes the potential energy in the $\Delta$p by importing protons from the IMS to the matrix to synthesize ATP (Mitchell, 1966). However, the mitochondrial matrix is still negatively charged in ATP-synthesizing or coupled mitochondria, as $\Delta$p consumption by the ATP synthase is the bottleneck and rate-limiting process controlling OXPHOS (Fig. 1A,C). In other words, the ETC has a higher capacity and activity producing $\Delta$p than ATP synthase consuming it, explaining why the matrix is negatively charged and has a basic pH despite ongoing ATP synthase-mediated proton import (see dials in Fig. 1A–D). The ATP/ADP exchangers (ANT) have a major contribution to the $\Delta$p consumption associated with ATP synthesis (Hafner et al, 1990). These exchangers import the ADP needed to produce ATP in the mitochondrial matrix, while exporting the ATP already synthesized. The 1:1 ATP/ADP exchange consumes 1 charge, equivalent to the import of 1 H+, as the matrix loses one negative charge because ADP has three negative charges and ATP has four. In this context, limiting matrix ADP availability is sufficient to stop the translocation of protons back into the matrix, as the ATP synthase and ANT will miss their key substrate needed for $\Delta$p consumption. Consequently, the decrease in $\Delta$p consumption caused by a limitation in ADP availability will increase the $\Delta$p (Nicholls and Ferguson 2013). This elevation in $\Delta$p will slow down electron transport and oxygen consumption: a domino effect explaining why the direct inhibition of ATP synthase by oligomycin induces an immediate decrease in oxygen consumption and a concurrent increase in $\Delta\Psi$m (Fig. 2A).

The fact that mitochondrial ATP synthase consumes $\Delta$p to generate ATP means that an increase in mitochondrial ATP synthesis and oxygen consumption can be associated with a decrease in $\Delta\Psi$m values (Fig. 1C vs. D), because $\Delta$p consumption to produce ATP might not be perfectly compensated by an elevation in electron transport. On the other hand, higher oxygen consumption can be associated with an increase in $\Delta\Psi$m as well, when the elevation in electron transfer rates is larger than the capacity of ATP synthase to produce ATP (see dials in Fig. 1A vs. B). An illustrative example is that pancreatic beta-cells increase mitochondrial respiration in response to an increase in extracellular glucose, not to meet cellular ATP needs and preserve the ATP/ADP ratio, but to generate signals stimulating insulin release based on glucose levels. Therefore, the increase in respiration induced by high glucose is not a response to an increase in matrix ADP as a result of an elevation in cellular ATP consumption. Indeed, the increase in respiration by high glucose hyperpolarizes beta-cell mitochondria and is associated with elevated ATP/ADP (Fig. 1A,B) (Lewandowski et al, 2020). In conclusion, these basic

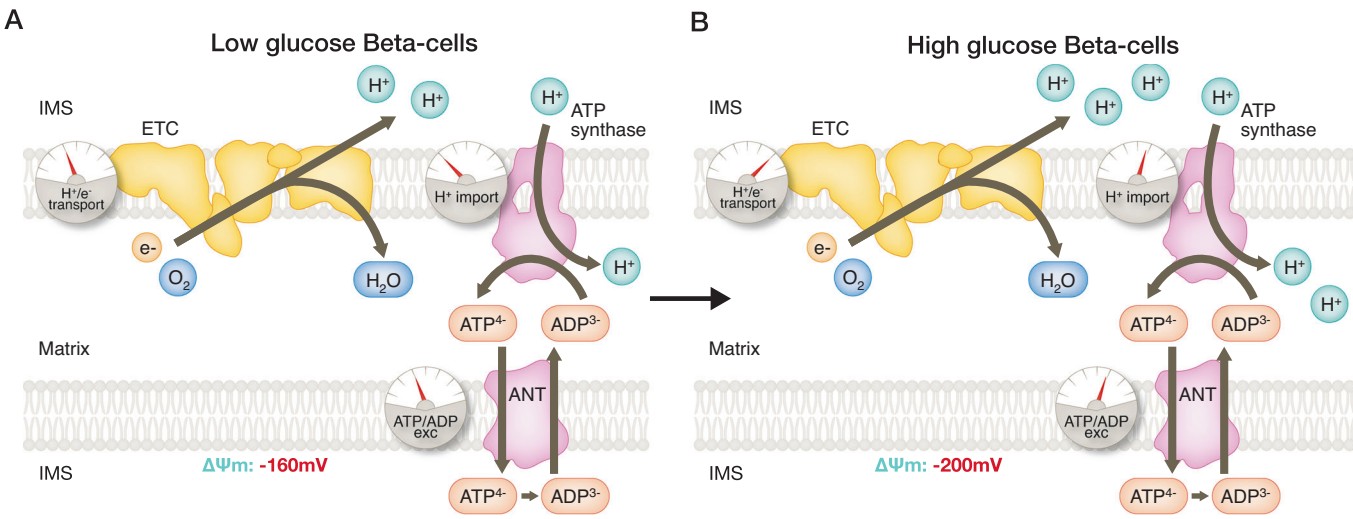

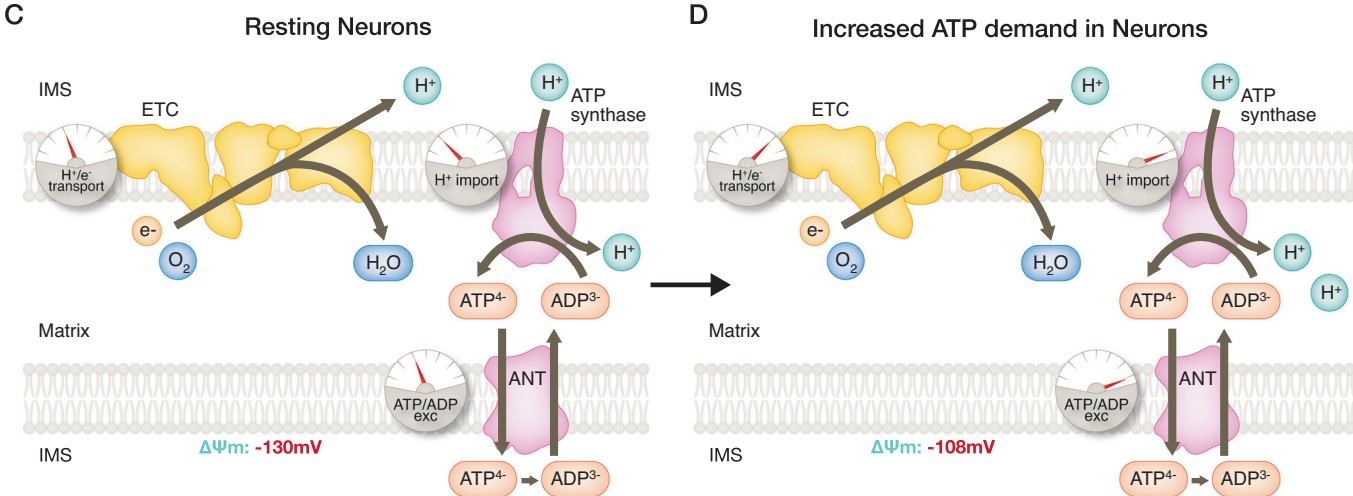

**Figure 1.  The relationship between membrane potential, respiration and ATP synthesis in coupled mitochondria.**

The two main modules, the electron transport chain (ETC) and ATP synthesis, are represented as cartoons from their structures. The membrane potential ($\Delta\Psi$m) is determined by the difference in charge and proton concentration across the inner membrane. The charge is determined by protons but also by ADP and ATP exchange. Protons do not accumulate, but they are in constant motion, as well as the ATP and ADP molecules. The dials represent the flux of the process per unit of time in each module and how they are connected by protons. (A, B) In beta cells, exposure to high glucose is sufficient to increase electron transfer and oxygen consumption. This increase in respiration is not activated by an increase in ATP demand, and thus it will hyperpolarize mitochondria, as the ATP synthase will act as a bottleneck. (C) In the basal state, electron transfer and oxygen consumption in neurons are limited by ATP synthesis-mediated $\Delta$p consumption. (D) When the cellular ATP demand increases and their ATP/ADP ratio decreases, mitochondrial ATP synthesis will be increased by ADP availability and the ETC will increase electron transfer and oxygen consumption per minute to counteract the drop in ATP/ADP.

principles of OXPHOS function show that measuring $\Delta\Psi$m is not sufficient to conclude on changes in OXPHOS in coupled mitochondria or in ATP demand by the cell. Thus, if other evidence suggests that mitochondrial function is changed despite the absence of a shift in $\Delta\Psi$m, oxygen consumption should be measured in addition, as it is a more sensitive parameter detecting changes in OXPHOS (see "$\Delta\Psi$m has low sensitivity and specificity reporting changes in OXPHOS activity in coupled mitochondria").

## $\Delta\Psi$m has low sensitivity and specificity, reporting changes in OXPHOS activity in coupled mitochondria

The inner membrane cannot support unlimited differences in charge between its matrix and the IMS, hence OXPHOS is inactive at low pH. An illustrative example is that oxygen consumption is halved when isolated mitochondria are incubated at pH 6.6 (Milliken et al, 2020). Accordingly, recent pH measurements in intact cells reported pH values of 7.9-8.0 in the mitochondrial matrix and 6.9-7.0 in the IMS (Rieger et al, 2021). On the other hand, if the $\Delta$p is eliminated, electron transport is impaired. This impairment is most likely explained by the need of the $\Delta\Psi$m to import the substrates across the inner membrane, whose oxidation in the matrix provides electrons to the ETC (Clerc and Polster, 2012). The inhibition of electron transport by the collapse of $\Delta$p is illustrated by the opposite effects on oxygen consumption of

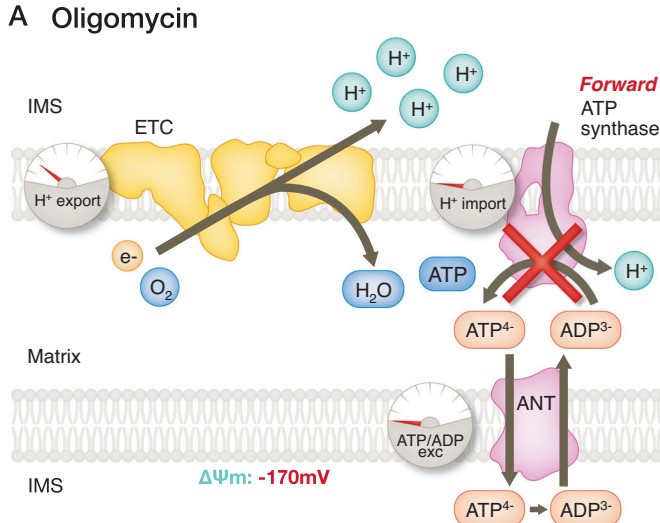

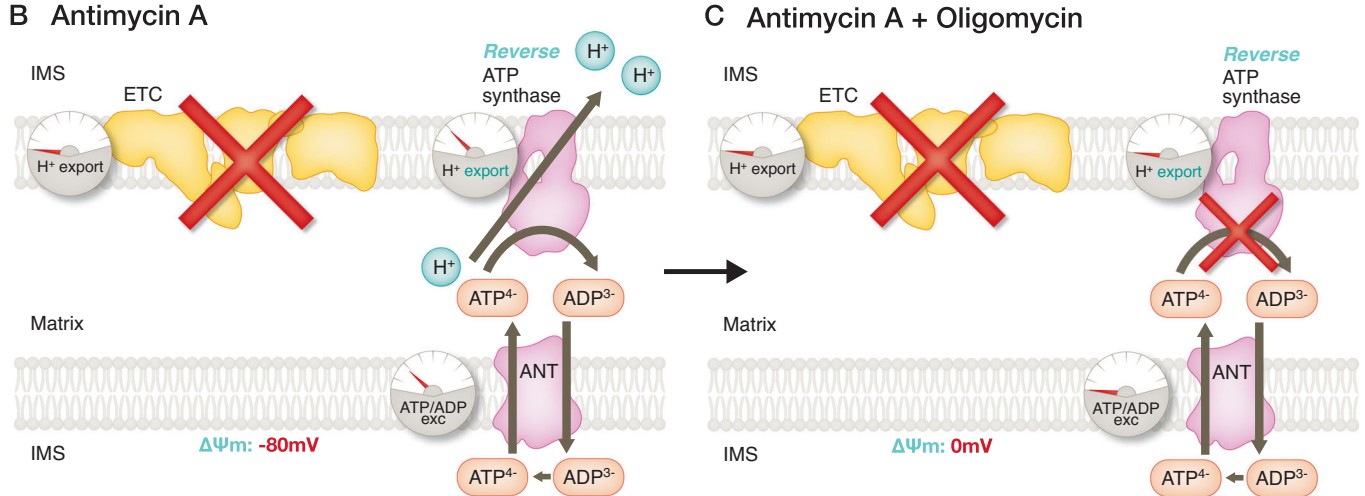

**Figure 2.  Mitochondrial ΔΨm enables discriminating between ATP-synthesizing and ATP-hydrolyzing mitochondria.**

(**A**) The blockage of ATP synthase by oligomycin in coupled mitochondria causes hyperpolarization. (**B**) Antimycin A, a complex III inhibitor, disrupts the ability of the ETC to generate ΔΨm and causes the reversal of ATP synthase, which then hydrolyzes ATP to generate a ΔΨm. (**C**) Adding oligomycin to antimycin A treated cells dissipates the ΔΨm, as it blocks ATP synthase reverse activity generating ΔΨm.

chemicals that promote proton entry, depending on their dose, i.e., protonophores like FCCP. While low concentrations of these protonophores will stimulate oxygen consumption, as they will increase proton import to rates similar to what the ATP synthase can import, higher protonophore concentrations in turn dissipate ΔΨm and decrease respiration (Clerc and Polster, 2012). Hence, upon Δp abrogation, the reverse mode of the ATP synthase can be activated to extrude protons and restore some Δp when the ETC collapses (Fig. 2B) (see next subchapter) (Nicholls and Lindberg, 1972). Altogether, these properties

reveal that there is a finite range of ΔΨm in coupled mitochondria, needed for the stability of the OXPHOS system.

In addition to this finite ΔΨm range, the normal mitochondrial response of the ETC to a decrease in Δp is to elevate electron transport and proton extrusion rates, as proton translocation is thermodynamically favored. This response of the ETC enables maintaining the Δp as close as possible to the resting state under conditions in which Δp consumption is increased. This response, increasing electron transport and oxygen consumption, preserving the Δp and ΔΨm, is essential for cellular bioenergetics.

A drop in Δp can be caused by an increase in ADP availability to decrease the ATP/ADP ratio, the latter being a sign that the cell needs their mitochondria to produce more ATP to support an activity or handle a stress. Thus, the ETC responds to changes in Δp consumption, trying to preserve this finite Δp range, at which the system is thermodynamically stable. This response further illustrates that ΔΨm is a parameter with a narrow dynamic range. A paradigmatic example is comparing the changes in oxygen consumption versus the changes in ΔΨm when ADP is added to isolated mitochondria: oxygen consumption can

increase up to 4-10-fold, while the maximum decrease detected in $\Delta\Psi m$ will only be around 10-25% (Kowaltowski and Abdulkader, 2024). While the exact $\Delta\Psi m$ range in mitochondria from intact cells remains unknown, measurements of the $\Delta\Psi m$ in intact neurons revealed a range between $-108$ and $-156$ mV under different physiological challenges (Gerencser et al, 2012), similar to what is observed in isolated mitochondria. Under resting conditions, $\Delta\Psi m$ in intact neurons was $-139$ mV, dropping to $-108$ mV under conditions of maximal cellular ATP consumption. Thus, a large change in OXPHOS rates in intact neurons is just inducing a 22% change in the $\Delta\Psi m$ as well (compare dials Fig. 1C vs D). A similar fold change is observed in pancreatic beta-cells stimulated with high glucose, which can increase $\Delta\Psi m$ from $-160$ to $-200$ mV (Gerencser et al, 2016), while doubling or tripling respiration (Shum et al, 2022) (compare dials Fig. 1A vs B).

Finally, it is to be noted that mitochondria also import and export *other* ions and charged metabolites across their inner membrane, which will change the $\Delta\Psi m$ to which the ETC will respond as well. Consequently, changes in $\Delta\Psi m$ and oxygen consumption will not always directly correlate with changes in mitochondrial ATP synthesis. Such a scenario would be revealed by an increase in basal respiration associated with a decrease in $\Delta\Psi m$, in the absence of changes in ATP-producing respiration. With such a result, researchers should then aim to investigate a specific change in the transport of additional charged metabolites and ions as the cause for the changes in $\Delta\Psi m$ and oxygen consumption, rather than changes in cellular ATP demand.

### $\Delta\Psi m$ measurements have high sensitivity in detecting uncoupled mitochondria, the absence of ETC activity and the reversal of mitochondrial ATP synthase

As mentioned above, not all mitochondria couple ETC to ATP synthesis and different drugs can promote an uncontrolled entry of protons to the matrix by distinct mechanisms. An illustrative example of physiological uncoupling is the activation of uncoupling protein 1 (UCP1) in brown adipose mitochondria to produce heat in response to cold. UCP1 activation can decrease the $\Delta p$ from 160 to 2 mV (Nicholls,

2023). Indeed, such a decrease in $\Delta p$ is incompatible with mitochondrial ATP synthesis, one reason being that the ATP/ADP exchanger will work in reverse mode at such low $\Delta p$, bringing ATP into the matrix instead of ADP (Brunetta et al, 2024; Chinopoulos et al, 2010). Another important example is the opening of the mitochondrial permeability transition pore (PTP) in the inner membrane, which can occur in mitochondria from many different cell types. This opening can be reversible or irreversible, hence the PTP can be activated to dissipate the $\Delta\Psi m$ transiently or permanently (Hüser and Blatter, 1999). An irreversible opening of the PTP causes the release of mitochondrial cytochrome c to kill cells by apoptosis. Thus, observing a large change in $\Delta\Psi m$ at resting state reveals the existence of uncoupling or a complete abrogation of ETC, a scenario incompatible with mitochondrial ATP synthesis activity by OXPHOS. For instance, if one detects a depolarization of 90 mV using dyes, but respiration assays still show oxygen consumption coupled to ATP synthesis, this means that something went wrong in the $\Delta\Psi m$ measurements (i.e., changes in fluorescence not driven by $\Delta\Psi m$). The best way to discern whether such a loss in $\Delta\Psi m$ represents uncoupling or the absence of ETC function is to measure oxygen consumption: UCP1 activation causes a large increase in oxygen consumption that produces heat instead of ATP, while ETC disruption and PTP opening cause a large decrease in oxygen consumption.

If an insufficient $\Delta p$ is present across the inner membrane, because the ETC is dysfunctional or even absent, the ATP synthase can also work in reverse mode: the ATP synthase will hydrolyze ATP to ADP and extrude protons to preserve the $\Delta p$ (Chinopoulos et al, 2010; Nicholls and Lindberg, 1972). The normal, forward, mode of ATP synthase produces ATP and imports protons to the matrix, thus hyperpolarizing when the ATP synthase is inhibited with the compound oligomycin (Fig. 2A). The ability of the ATP synthase to shift from the forward mode to work in reverse shows that the $\Delta p$ can be key for other biosynthetic functions occurring in the mitochondria, different to OXPHOS. As a result, ATP molecules are diverted away from other vital processes into the mitochondrial matrix to fuel the ATP synthase in reverse mode and preserve the $\Delta p$. In this case, $\Delta\Psi m$ measurements are key to identify

this reverse activity, as reversal causes a qualitative difference in $\Delta\Psi m$ responses to oligomycin treatments: ATP-hydrolyzing mitochondria with their ETC abrogated will depolarize or lose $\Delta\Psi m$ when treated with oligomycin (Fig. 2B vs. C), which is the opposite of what is observed in coupled mitochondria (Fig. 2A). The reason is that oligomycin blocks the ability of ATP synthase to carry protons in both directions. Thus, in some cases, oligomycin treatments are needed to determine whether a change in $\Delta\Psi m$, such as the one induced by complex III inhibition with Antimycin A (Fig. 2B vs. C), might represent mitochondria having their ATP synthase working in the reverse mode. In this regard, we recently identified that, in some pathologies, the excessive activation of the reverse mode of the ATP synthase that hydrolyzes ATP is maladaptive (Acin-Perez et al, 2023).

### The second hallmark: specificity and sensitivity reporting $\Delta\Psi m$ varies between different dyes, but even diverges using the same dye in different metabolic states and cells

Mitochondrial dyes are fluorescent compounds that *preferentially* concentrate and accumulate inside mitochondria in live cells, enabling their visualization and analysis through fluorescence microscopy or flow cytometry (Ehrenberg et al, 1988). These dyes do not stain mitochondria as antibodies do, as they do not specifically and exclusively recognize a single mitochondrial protein or lipid. Instead, these probes preferentially accumulate inside the mitochondria because of two chemical properties: they can cross biological membranes and they are cations that are drawn to the negatively charged matrix-side of the inner membrane (Desai et al, 2024; Ehrenberg et al, 1988; Wolf et al, 2019). These properties enable that dye distribution into the mitochondria will follow the Nernst equation and therefore, dye fluorescence can be used to determine $\Delta\Psi m$. However, some dyes show additional properties breaking the correlation between dye fluorescence and $\Delta\Psi m$, which causes the dye to behave in a non-Nernstian manner. In other words, Nernstian behavior means that the fluorescent staining of mitochondria by dyes will respond in real time to changes in $\Delta\Psi m$, while fluorescent staining determined by other processes will not respond (Desai et al, 2024; Ehrenberg et al, 1988;

**Table 1.** List of dyes and concentrations: the concentration ranges vary with the cell type, their state and use across laboratories, depending on detection methods.

| | TMRM/TMRE | Rhod123 | JC-1 | MTG | MTDR |
|---|---|---|---|---|---|
| Quenching kinetic ΔΨm measurements | Live cells 0.1–25 μM (Esteras et al, 2020; Perry et al, 2011) | Preferred Live cells 0.1–10 μM (Emaus et al, 1986; Perry et al, 2011) | Live cells binary assessment of polarized state (complete ΔΨm dissipation) 0.25–8 μM (Perry et al, 2011; Wolosin et al., 2017) | Live cells flow cytometry + imaging: Mitochondrial mass, morphology, and normalized TMRE in cells with some polarization 50 nM–5 μM (Desai et al, 2024) | Live and fixed cells, flow cytometry + imaging and homogenates: Mitochondrial mass and morphology 50 nm–1 μM (Acin-Perez et al, 2020; Desai et al, 2024) |
| Non-quenching Steady-state ΔΨm and for kinetic analyses | Preferred Live cells 10–40 nM (Esteras et al, 2020) | Live cells 10–40 nM (Esteras et al, 2020) | | | |
| Working concentrations | 15–500 nM (HeLa, myoblasts, COS7), 15–200 nM (MEFs) | 5 μM (HeLa, myoblasts) | 10 nM (myocytes, PC12), 2 nM (neurons), 0.3 nM (human fibroblasts) | 200 nM (HeLa, MEFs, myoblasts) | 500 nM, 50 nM (human fibroblasts) (Desai et al, 2024) |

Thus, validation experiments and controls are needed for each model and cell type.

Wolf et al, 2019). As a result, the best approach to determine whether a dye and which fraction of the dye is reporting on ΔΨm is to induce acute and specific changes in mitochondrial ΔΨm and establish whether the fluorescent staining of mitochondria responds, in real time, to the induction. The most common approach is to treat mitochondria with oligomycin and determine whether the fluorescence of the dye increases, and whether treatment with protonophores that depolarize mitochondria eliminates the fluorescence of the dye inside mitochondria. Any fluorescence left inside mitochondria after ΔΨm depolarization will be considered as non-Nernstian. Consequently, a key consideration is the use of the *minimal concentrations* of any of these dyes to: (i) diminish the number of dye molecules with non-Nernstian behavior, (ii) prevent toxicity after laser exposure, and iii) prevent the decrease in fluorescence emission when the concentration of a dye is too high, a process known as quenching (see Section 3). The following list of dyes is meant to give an overview of available options and their respective best use scenarios (Table 1).

**Nernstian-mitochondrial dyes**

Dyes for which a Nernstian behavior of the majority of molecules can be achieved following the correct protocols.

***Tetramethyl-rhodamine ethyl and methyl esters (TMRE and TMRM)***
These are the most used dyes, which respond with high sensitivity to changes in ΔΨm and for which the non-Nernstian behavior can be easily mitigated. These dyes can cross the plasma and mitochondrial membranes effectively, to concentrate largely inside the mitochondria with relatively low binding to mitochondrial lipids and proteins. Consequently, most of their fluorescence detected in cells with polarized mitochondria stems from the accumulation of the dye in the mitochondria, and its fluorescence has a Nernstian behavior. However, as with other dyes, TMRE or TMRM concentrations need to be optimized, and ideally employed in the lower concentration range. The published concentrations and incubation times of TMRE/TMRM that enable diffusion and proper equilibration across membranes, but without enhancing laser-induced toxicity and non-Nernstian behavior, differ substantially across laboratories (e.g., TMRE at 15 nM for 1 h in Wolf et al, 2019 vs 500 nM for 10 min in Kleele et al., 2021 in HeLa cells). For these reasons, we strongly suggest to optimize concentrations and time of dye incubation before image acquisition. The concentration is the most important aspect, as TMRE/TMRM bound to mitochondrial lipids show a 100-fold increase in fluorescence when compared to free TMRE/TMRM, meaning that TMRE/TMRM molecules stuck to lipids that can be ΔΨm insensitive can have a greater contribution to the fluorescence detected than the one stemming from free molecules. In addition, the use of high TMRE/TMRM concentrations amplifies laser-induced toxicity and can cause the dye to be in the quenching mode (see section "The third hallmark: dye concentrations determine whether ΔΨm depolarization is reported by an increase or a decrease in dye fluorescence (quenching versus non-quenching modes)"). While the TMRE/TMRM-based ratiometric approach was shown to improve the specificity of TMRE fluorescence reporting on ΔΨm in isolated mitochondria, this technique is problematic in intact cells and tissues like the perfused heart (Scaduto and Grotyohann 1999; Fisher-Wellman et al, 2018).

***Rhodamine 123***
TMRM and TMRE are variants of Rhodamine 123 (Rhod-123), which is more hydrophilic and crosses membranes only slowly. It is a dye that is particularly useful for ΔΨm measurements in the quenching mode (see section "The third hallmark: dye concentrations determine whether ΔΨm depolarization is reported by an increase or a decrease in dye fluorescence (quenching versus non-quenching modes)").

***Safranin O***
Does not cross cellular membranes easily, but crosses mitochondrial membranes. As a result, Safranin O is mostly used in assays using isolated mitochondria rather than intact cells.

***JC-1(5,5',6,6'-Tetrachloro-1,1',3,3'-tetraethylbenzimidazolyl-carbocyanine iodide)***
This dye aggregates and emits red fluorescence from mitochondria when ΔΨm is high. On the other hand, JC-1 is found in a monomeric form that emits green fluorescence when ΔΨm is low. Thus, JC-1 could be technically considered as a Nernstian dye, as the formation of red aggregates can be reversible and ΔΨm dependent (Reers et al, 1995; Smiley et al, 1991). This qualitative property made JC-1 an attractive dye to measure mitochondrial heterogeneity in live cells, as hyperpolarized mitochondria stained with JC-1 will emit red fluorescence and depolarized mitochondria will emit green fluorescence. However, multiple

**Figure 3. MitoTracker Green (MTG) labeling can decrease over time upon cumulative laser exposure.**

Scheme showing examples of when MTG staining is not stable in cells being imaged by confocal microscopy. This lack of stability limits the specificity and accuracy of the TMRE/MTG ratio reporting on ΔΨm over time. Such a behavior can be explained by laser-induced phototoxicity, which can be limited by lowering the concentrations of the dyes, laser intensity, and time of laser exposure.

issues were identified that affect the reliability of JC-1. It also forms crystals outside the mitochondria, as well as irreversible red aggregates that specifically accumulate inside the nucleus of dead cells (Duchen et al, 2003), which are unevenly distributed inside individual mitochondria in a seemingly random non-cristae-like pattern (Smiley et al, 1991). Also, the fluorescence of the monomer can be just increased in a lipidic environment (Duchen et al, 2003), and the exact range of JC-1 concentrations that support the formation red aggregates specifically in hyperpolarized mitochondria remains unclear, as well as the concentrations at which changes in JC-1 green fluorescence report on ΔΨm with sensitivity and specificity (Perry et al, 2011). Finally, studies with more success showed that red aggregates can be dispersed to JC-1 green monomers at values below −140 mV (Keil et al, 2011; Reers et al, 1995; Smiley et al, 1991). This means that the red-to-green ratio cannot be used to differentiate a mitochondrion in the resting state, with ΔΨm = −140 and −130 mV, versus a mitochondrion with increased ATP synthesis with ΔΨm = −115 and −108 mV (Fig. 1C,D) (Perry et al, 2011). Thus, it seems that changes in red-to-green fluorescence JC-1 ratio might only be able to report large changes in ΔΨm, such as the ones occurring when ETC activity is completely abrogated. We can conclude that JC-1 is a dye with lower sensitivity, specificity and reliability to measure changes in ΔΨm, when compared to other dyes.

### Non-Nernstian-mitochondrial dyes

The majority of these dyes are used to determine mitochondrial mass and morphology, as their non-Nernstian labeling of mitochondria disables their use as sensitive and accurate ΔΨm reporters. We will cover the most widely used:

### MitoTracker Green (MTG)

It binds to mitochondrial proteins in a covalent manner, as it has chloromethyl groups that can react with free thiol groups of cysteine residues, presumably via alkylation (Presley et al, 2003). MTG covalent binding is not as toxic as MitoTracker Orange, which can induce mitochondrial dysfunction and mitochondrial pore opening (Scorrano et al, 1999). This covalent binding is the key feature that allows MTG to be retained within the mitochondria even after ΔΨm loss (Desai et al, 2024; Wikstrom et al, 2014). When analyzing ΔΨm in individual mitochondria by confocal microscopy, MTG enables correcting differences in fluorescence caused by different positions of the mitochondria in the Z axis, which impact the detection of their real fluorescence. Thus, MTG is widely used to perform measurements of mitochondrial mass and morphology, as well as to normalize TMRE/TMRM signal. However, MTG is a positively charged lipophilic cation as well, which means that it needs mitochondria with a Δp to be able to stain mitochondria. Thanks to the covalent binding of MTG to mitochondrial proteins, though, the accumulation of MTG inside mitochondria over sufficient loading time can mask any initial differences in loading caused by an initial difference in ΔΨm between groups. Thus, while MTG stays in mitochondria after their depolarization and is considered membrane-potential independent, MTG will not be able to accumulate with high efficiency in completely depolarized mitochondria. An illustrative example is that adding MTG to cells with their mitochondria completely depolarized by protonophore treatment will not be effective in labeling mitochondria. Such Δp dependency combined with binding means that if an excess of MTG molecules that did not bind to mitochondrial proteins is not properly washed, or if changes in the mitochondrial redox balance occur that break the expected thioether bond of MTG to mitochondrial proteins, such events can decrease the MTG signal in mitochondria over time, particularly after laser exposure. In Fig. 3, we show a scheme illustrating how the MTG signal can be lost with cumulative laser exposure and that this decrease can be accelerated in cells exposed to mitochondrial stress. These

examples showcase the need of properly controlling whether MTG labeling is stable for its use as a normalization dye.

### MitoTracker Deep Red (MTDR)

It is a variant of MTG, but with a lower number of chloromethyl groups, among other differences. Unlike MTG, MTDR fluorescence is preserved after chemical fixation (Clutton et al, 2019; Cottet-Rousselle et al, 2011). In addition, and by a mechanism to be determined, MTDR can still preferentially accumulate in already depolarized mitochondria. Thus, MTDR has been uniquely useful to quantify mitochondrial mass in tissue homogenates (Acin-Perez et al, 2020).

### Mitochondrially-targeted fluorescent proteins

Mitochondrial mass can also be assessed by using mitochondrially targeted fluorescent proteins, such as GFP and DsRed. Notably, the specificity and sensitivity of this approach to quantify mitochondrial mass is much larger when compared to mitochondrial dyes. Some protein reporters for $\Delta\Psi$m have been described in yeast that enable concurrent detection of mitochondrial morphology and membrane potential, the latter based on the requirement of a $\Delta\Psi$m to import proteins into the mitochondrial matrix (Vowinckel et al, 2015). However, these approaches tend to be less popular, as they require transfection, transduction with viral vectors or generating transgenic organisms. In addition, the excessive overexpression of these artificial proteins can cause proteotoxicity in the mitochondrial matrix. Therefore, despite being more advantageous in terms of sensitivity and specificity, the need for transgenic expression discourages their usage in mitochondrial $\Delta\Psi$m measures.

### The specificity and sensitivity of a $\Delta\Psi$m-reporting dye reporting can vary with the cell type and their state

Multiple cancer cell lines, such as Hep-2 (Diaz et al, 2001) and hematopoietic stem cells (Almeida et al, 2017), express ATP-dependent Multidrug resistance (MDR) transporters that have evolved to expel xenobiotics. This mechanism is not only present at basal conditions in the aforementioned cell types, but it can also be induced in cells exposed to stress, allowing

them to "spit out" these dyes. As a result, elevated MDR activity can be misinterpreted as mitochondrial depolarization, as the dye will be unable to reach and stain mitochondria. One of the best-known proteins involved in MDR is P-glycoprotein or ABCB1, which tends to decrease the intracellular concentration of its substrates. Indeed, Rhod-123 was used in the past as a dye to track ABCB1 activity, rather than to track mitochondrial $\Delta\Psi$m (Gottesman and Pastan, 1993). Consequently, MDR must be inhibited to measure $\Delta\Psi$m when using these dyes. The use of the calcium channel blocker Verapamil, an inhibitor of these transporters, can be employed to overcome this issue (Morganti et al, 2019).

## The third hallmark: dye concentrations determine whether $\Delta\Psi$m depolarization is reported by an increase or a decrease in dye fluorescence (quenching versus non-quenching modes)

Aggregation-induced quenching (see Glossary) is a phenomenon in which the accumulation of molecules of the same probe mitigates their own fluorescence. In this case, high concentrations of the mitochondrial dyes reporting on $\Delta\Psi$m induce their aggregation in a manner that induces quenching: the adjacent molecules in the aggregate result in the absorption of photons emitted, thereby reducing the amount of fluorescence emitted with increasing concentrations (Duchen et al, 2003; Emaus et al, 1986). As a result, the mitochondrial dyes discussed can report changes in $\Delta\Psi$m in these two different modes:

### The non-quenching mode

TMRM/TMRE or Rhod-123 used at nanomolar concentrations will accumulate and fluoresce inside the mitochondria as a function of $\Delta\Psi$m, and thus will work in a non-quenching mode. In other words, an increase in fluorescence will be observed when mitochondria hyperpolarize and a decrease in fluorescence when they depolarize. The linearity of the fluorescence enables the calibration of membrane potential through the introduction of known quantities of potassium in the presence of valinomycin and calculating values from the Nernst equation (Ehrenberg et al, 1988).

However, a change in the aqueous content of the medium induced by this approach may induce a shift in the spectral behavior of the probe, which needs to be taken into account (Kowaltowski and Abdulkader, 2024).

### The quenching mode

When working with TMRM/TMRE or Rhod-123 at elevated concentrations (micromolar, or even high nanomolar), the process of auto-quenching is significantly favored (Duchen et al, 2003). In such conditions, upon depolarization, some molecules of the dye will translocate from the mitochondria to the cytosol, which will cause dye dequenching to increase fluorescence initially inside the mitochondria, and in the cytosol by the molecules that translocated from the mitochondria. On the other hand, mitochondrial hyperpolarization will increase dye concentration inside mitochondria to promote aggregation and auto-quenching, causing a drop in fluorescence (Perry et al, 2011). The fact that dye fluorescence inside the mitochondria did not scale linearly with quenching concentrations of TMRM/TMRE and Rhod-123, disables their proper calibration and their use in the quenching mode to determine steady-state $\Delta\Psi$m (Emaus et al, 1986; Esteras et al, 2020). As a result, the quenching mode is helpful to identify manipulations depolarizing mitochondria.

### Validation of the $\Delta\Psi$m-reporting mode and biological variables affecting quenching

Given that other biological variables different than the mitochondrial $\Delta\Psi$m determine how much dye gets into the mitochondria and the cytosol, it is imperative to ascertain the mode and state of the dye (Kowaltowski and Abdulkader, 2024). This means that in any $\Delta\Psi$m measurement, the addition of a mitochondrial protonophore, such as FCCP, as a control is needed. The addition of an uncoupler will induce an increase in fluorescence in the quenching mode, and a decrease in the non-quenching mode (Esteras et al, 2020). It is important to remark that measurements monitoring mitochondrial mass, cell death and MDR activity (i.e., dye retention by the cell and permeability) should be performed when using the quenching mode, as these differences might change the concentrations needed to achieve quenching. Consequently,

## Flow cytometry feasibility flow chart

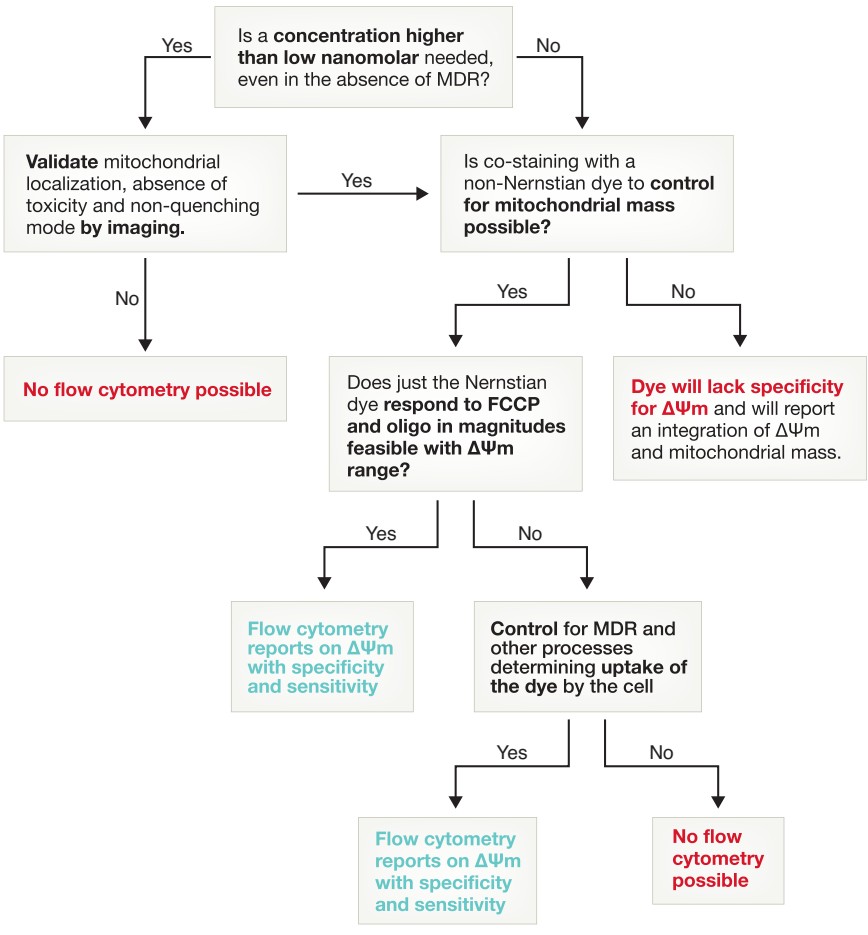

**Figure 4. Decision tree on how to ensure an adequate measurement of ΔΨm by flow cytometry.**

Main parameters to control and when imaging is needed to validate the specificity and sensitivity of using flow cytometry to measure the fluorescence of dyes reporting on ΔΨm.

depending on the concentration of the dyes used and cell state, mitochondrial depolarization can be observed as an increase or a decrease in TMRM/TMRE or Rhod-123 fluorescence.

### The fourth hallmark: imaging is needed to ensure the specificity and accuracy of flow cytometry-based ΔΨm and ROS measurements

Two of the most used approaches to quantify ΔΨm and ROS generation in the mitochondria using fluorescent dyes are flow cytometry and live-cell imaging. Microscopy tends to provide a more reliable assessment of mitochondrial parameters, as imaging enables: (i) to determine the fluorescence emitted from the mitochondria and (ii) to measure fluorescence intensity

per mitochondrion. As mentioned, Nernstian dyes do not exclusively bind and fluoresce inside mitochondria. They accumulate more in mitochondria, but these dyes will also fluoresce and accumulate in the cytosol, particularly when mitochondria are depolarized.

On the other hand, flow cytometry only provides fluorescence intensity per cell. As a result, cells that show an increase in mitochondrial mass or in plasma membrane polarization will show higher fluorescence of ΔΨm-sensitive dyes by flow cytometry, even in the absence of changes in mitochondrial ΔΨm. Thus, measurements of ΔΨm by flow cytometry require the use of MTG to monitor and normalize for mitochondrial mass (Clutton et al, 2019; Morganti et al, 2019), as well as of mitochondrial protonophores to determine

how much of the fluorescence follows a Nernstian behavior and reports on mitochondrial polarization (Morganti et al, 2019). While MTDR could potentially be better to determine mitochondrial mass also by flow cytometry, its excitation and emission spectra have a significant overlap with most ΔΨm-sensitive dyes. In addition, higher concentrations of the dye might be needed to detect its fluorescence by flow cytometry. These limitations of flow cytometry become particularly relevant when employing other dyes, like MitoSOX, which reports the levels of an ETC byproduct, superoxide. High concentrations of MitoSOX induce its preferential translocation and accumulation in the nucleus (Johnson-Cadwell et al, 2007), it can affect mitochondrial respiration (Roelofs et al, 2015), and even report on other species different than superoxide. These undesired effects are seen both at high concentrations and extended incubation times. Hence, without proper controls, a flow cytometry assay with these probes will likely result in misleading interpretations. In this case, live imaging serves as a key complementary tool to confirm proper mitochondrial localization and absence of mitochondrial stress, the latter being revealed by preserved morphology at the probe concentrations needed for flow cytometry (Fig. 4).

Imaging also allows for the detection of differences between mitochondria within a cell: peridroplet versus cytosolic mitochondria or even between individual cristae in a single mitochondrion (Benador et al, 2018; Wolf et al, 2019). This further highlights the importance of microscopic observation of ΔΨm, since such events cannot be observed or quantified by flow cytometry. Moreover, imaging is compatible with time-lapse experiments, essential for studying dynamic and reversible ΔΨm changes to specific treatments. While imaging can be more labor-intensive and lower throughput, it provides qualitative and quantitative insights that flow cytometry cannot match.

There are several ways to validate whether flow cytometry can provide an accurate assessment of ΔΨm: the most important ones that must be incorporated to obtain reliable results are positive controls modulating ΔΨm, such as FCCP and oligomycin treatments to decrease and increase mitochondrial ΔΨm respectively, (Perry et al, 2011), and negative controls, such as unlabeled cells. One of the most relevant experiments, however, is to validate by imaging that the concentrations of

dye used by flow cytometry show fluorescence confined to the mitochondria, do not affect mitochondrial morphology and report the $\Delta\Psi$m in the expected mode (i.e. quenching vs non-quenching, see Section 3). A decision-making guide summarizing when flow cytometry can be used is presented in Fig. 4.

## Conclusions

Mitochondrial membrane potential measurements using fluorescent dyes in intact cells are one of the most accessible approaches to determine mitochondrial bioenergetic function. However, its assessment is subject to multiple technical and biological variables that need to be controlled. Moreover, the same accurately measured shift in $\Delta\Psi$m can be reflecting opposite changes in mitochondrial OXPHOS, meaning that additional complementary approaches are needed to draw conclusions on OXPHOS. We also hope to illustrate that mitochondrial OXPHOS can even change in the absence of detectable changes in $\Delta\Psi$m and vice-versa, and that the lack of correlation between OXPHOS and $\Delta\Psi$m can be explained by the narrow $\Delta\Psi$m range at which coupled mitochondria function, as well as other biological factors determining dye uptake and their non-Nernstian behavior. In this piece, we focused on how to reliably determine changes in $\Delta\Psi$m between different groups or conditions using intact cells. However, for procedures to obtain absolute measurements of mitochondrial $\Delta\Psi$m, we refer readers to (Gerencser et al, 2012), in which details how other key parameters such as mitochondria volume and condensation, cell volume, and viscosity need to be controlled and modelled. We also want to acknowledge that isolated mitochondria and permeabilized cells can be used to monitor $\Delta\Psi$m, and we refer readers to these publications (Scaduto and Grotyohann 1999; Fisher-Wellman et al, 2018) for extended reading. We hope that this piece will facilitate a more accurate and consistent assessment of mitochondrial function in laboratories of diverse backgrounds, established or beginners alike.

## Key considerations

- **Validate that the fluorescence of the dye follows a Nernstian behavior:** treatment with a mitochondrial protonophore will change its fluorescence in the opposite direction compared to a mitochondrial ATP synthase inhibitor.
- **Validate that the used dye concentration provides the intended mode:** non-quenching versus quenching mode and mostly enriching its fluorescent signal in mitochondria. Different cell types and different stress might activate multidrug resistance activity or change membrane permeability to the dyes. Thus, it is not obvious that the same dye concentration will be in the same mode in different cell types and conditions. Treatment with a protonophore will inform on the mode.
- **Use multiple complementary approaches to determine the change in membrane potential:** this will ensure acquisition of a reliable readout for mitochondrial respiratory function and ATP synthesis. Modulation of ATP synthesis rates will cause a small change in membrane potential, either increasing or decreasing it. Large changes decreasing membrane potential will reflect severe electron transport chain dysfunction or uncoupling. Treatment with oligomycin inducing depolarization can reveal the presence of ATP-hydrolyzing mitochondria.
- **Prefer imaging over flow cytometry** for quantifying fluorescence of mitochondrial membrane potential and ROS-sensitive dyes. This will enhance specificity and accuracy of the measurements.

## Peer review information

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

## Acknowledgements

We would like to acknowledge the support by PID2021-127278NB-I00 from MCIN/AEI/ https://doi.org/10.13039/501100011033 and by "ERDF A way of making Europe", by the European Union, as well as by Fundación BBVA, Beca Leonardo LEO22-2-1659-BBM-BIO-18. We acknowledge support from the Scientific Network Enfermedades Metabólicas funded by the Consejo Superior de Investigaciones Científicas (CSIC), Spain.

## Author contributions

**Oscar Tovar-Ferrero**: Visualization; Writing—original draft; Writing—review and editing. **Javier Rubio**: Visualization; Writing—original draft; Writing—review and editing. **Antonio Zorzano**: Conceptualization; Writing—review and editing. **Guillermo Martinez-Corrales**: Visualization; Writing—original draft; Writing—review and editing. **Marc Liesa**: Conceptualization; Supervision; Funding acquisition; Visualization; Writing—original draft; Writing—review and editing.

## Disclosure and competing interests statement

ML is a co-founder of Enspire Bio LLC, was a consultant for Capacity Bio and received sponsored research agreements from Bantam Pharmaceutical, Janssen, Amgen and Senergy Bio.

