## [Peer Review File · The EMBO Journal]

Measuring mitochondrial membrane potential

Oscar Tovar-Ferrero, Javier Rubio, Antonio Zorzano, Guillermo Martinez-Corrales, and Marc Liesa

Corresponding authors: Marc Liesa (mlrbmc@ibmb.csic.es) , Guillermo Martinez-Corrales (gmcbbc@ibmb.csic.es)

Review Timeline:

Submission Date:	2nd May 25
Editorial Decision:	24th Jun 25
Revision Received:	10th Sep 25
Editorial Decision:	9th Oct 25
Revision Received:	21st Oct 25
Accepted:	22nd Oct 25

Editor: Daniel Klimmeck

Transaction Report:

Dear Marc,

Thank you again for sending us your commentary article for the metabolism advice series, as well as for your patience with our feedback. As mentioned, we have asked three dedicated experts to assess your manuscript, and in the meantime got feedback from all of them, which I enclose below.

As you will see, the colleagues much appreciate the commentary and find it highly valuable and worth publishing. They also provide constructive feedback on how to further improve it by amending technical aspects, complementing the discussion and also better acknowledge our broader audience by introducing basic concepts and terminology. I suggest adding a concise glossary for this purpose should be helpful. Further, the experts suggest a number of enhancements of the figure, which seem sensible to me. They also comment on title terminology and wording used.

With respect to the artwork, please consider adjustments and let me know about your views about the referee's point. In any case, a preliminary next version of the sketch will be sufficient as a basis for final rendering by our scientific illustration team.

I hope you will find the comments helpful. I am sure that a next version incorporating the suggestions made by the referees will be highly noted and appreciated. I would thus like to invite you to submit such a revised version using the link enclosed below.

Please let me know in case I can be of any help with this.

with
Best wishes,

Daniel

Daniel Klimmeck, PhD
Senior Editor
The EMBO Journal

Referee #1:

This is a detailed methods commentary about the pitfalls of assays for monitoring mitochondrial membrane potential, which will be very useful for any researchers interested in integrating these assays into their work. The authors comprehensively provide guidance for choosing and establishing appropriate assays, which should help improve rigor and interpretability in mitochondrial research. I only have a few suggestions the authors might want to consider for further improving the clarity of the manuscript.

1. For the broad readership of EMBO Journal, it is important to make sure that all relevant terms and concepts are explained. E.g. it would be helpful to define 'Nernstian behavior' and 'uncoupled mitochondria' when these terms are first introduced.
2. For assessment of mitochondrial mass, would it be helpful to include a brief discussion of mitochondrially targeted fluorescent proteins as organelle markers?
3. Figure 1 is critical for illustrating the commentary's concepts, but this figure is rather difficult to comprehend. I assume the arrows indicate charge flows? It could be helpful to code positive/negative charges with different colors. The text and arrows above the panels seem redundant with the charge flow depicted in the panels - consider to remove text and improve clarity of

these processes in the figure panels.

4. It might help to split Figure 1 into two figures: 1) Different physiological contexts. Here I find the cell types rather specialized info, consider emphasizing physiological state rather than cell type. 2) Experimental manipulations: This of course includes oligomycin, but it would be very helpful to illustrate other manipulations that are discussed in the text and how they inform on mitochondrial state, e.g. drugs changing potential, proton gradient, ATP synthesis, ATP/ADP antiport.

Referee #2:

Mitochondrial membrane potential is frequently used as the sole measure of mitochondrial metabolism and apoptosis, and the measurements are often inadequately performed or interpreted. Therefore, an authoritative review on this topic is much needed, strongly justifying the efforts of Tovar-Ferrero et al. The piece aims to provide 1) some bioenergetics basics to help guide interpretations including why measuring $\Delta\psi$ alone might not be enough to understand bioenergetics status, and 2) important details about commonly used tools to measure $\Delta\psi$, with the intention of minimizing misinterpretations. In my opinion, the ms needs improvements before it can be considered for publication.

Wild card refers to "uncertainty" whereas the issue often seems to be more one of difficulties in interpretation. Maybe the title/some subtitles would need reconsidering.

There needs to be a clear statement near the start of the section 1. that $\Delta\psi$ reflects a balance of polarization (H^+ from matrix to IMS side of IMM) and depolarization (H^+ from IMS to matrix side of IMM), and that multiple processes contribute to each (mainly the respiratory complexes pump H^+ from matrix to IMS) and several process move H^+ from IMS to matrix (e.g., ATP synthase, PiC, MPC).

In the section about the dynamic range of $\Delta\psi$, it should be stated that the actual dynamic range in mitos in intact cells is not really known. The Authors try to make the point that the $\Delta\psi$ will generally change only within a narrow range; they could add the caveat that $\Delta\psi$ would dissipate with PTP opening, sustained or flickering.

Most of the classic and many of the current $\Delta\psi$ measurements are carried out in isolated mitochondria or permeabilized cell suspensions in fluorometer cuvettes. I think this approach has to be described and discussed.

2.2.2. Based on the experience of this reviewer and other scientists, MitoTracker green fluorescence is not completely insensitive to $\Delta\psi$ upon accumulation into the mitochondria. This is another limitation in its use as $\Delta\psi$ -insensitive reference.

The ratio fluorescence approach for TMRE/TMRM/R123 has to be described in the review (PMID: 9876159; PMID: 30257218) Also, mentioning that TMRE isn't exactly nernstian and why is needed. This reviewer would swap sections 3 and 4

Minor

-Text refers to Fig1 A, B panels but no panel labeling is shown on Fig1.
Furthermore, Figure 1 top panels are not illustrative of what's trying to be explained.

For left panel, should show a much fatter arrow for H^+ matrix \rightarrow IMS, then, in right panel, the arrow showing H^+ IMS \rightarrow matrix should fatten

Adding a "dial" scheme to show "net" $\Delta\psi$ (stead state $\Delta\psi$) would be helpful: highly polarized for substrate only, then slightly depol for a little ADP, then more depol with more ADP

-The means of mitochondrial membrane Pg1 last para

The transfer of electrons across complexes I, III and IV requires proton translocation across the inner membrane. To correct content, please replace requires with "drives"

-Pg3 para1

higher protonophore concentrations that completely dissipate Ψ_m decrease respiration
delete "that"

-pg3 para 3

In an oxygen consumption experiment, this could be seen as increased basal and proton-leak driven oxygen consumption rates, without observing differences in absolute oligomycin-sensitive respiration rates
Not very clear

-pg4 para2

ATP synthase might be activated in excess in some pathologies, inducing ATP wasting
is it really meant that the pathology is "ATP wasting"?

Referee #3:

The central role of mitochondria in the signaling and metabolic processes of a eukaryotic cell have led to an increasing interest in mitochondrial bioenergetics. At the core of mitochondrial bioenergetics is mitochondrial membrane potential (MMP), the voltage differences across the inner mitochondrial membrane. The manuscript by Tovar-Ferrero et al provides a manual that will guide mitochondrial researchers in the appropriate tools and interpretation of commercially available fluorescent dyes that enable MMP measurements. The manuscript is timely and clear, and will be of great use to the community. The authors might consider some changes for clarity and utility.

Minor:

- 1) Organization. The authors might consider restructuring their manuscript with a first section 1 dedicated to first explaining mitochondrial OXPHOS basics. This might help all readers know the fundamentals, thereby enabling them to better understand the details of the relationship between OXPHOS and MMP. Then, a section 2 dedicated to understanding how to use and interpret commercially available dyes.
- 2) It is unclear why/what the authors are referring to with 'wild card.' A wild card refers to a person or thing whose influence is unpredictable and whose qualities are uncertain. As the authors discuss, the dyes are not so much unpredictable as incorrectly used. The authors might consider a simpler title, for example 'A guide to using membrane potential assays.' The continuous references to 'wild cards' are confusing and unclear. The authors might use 'considerations' rather than wild-cards.
- 3) The authors should more clearly explain proton extrusion and proton import.
- 4) The authors often make statements that make assumptions regarding prevailing beliefs in the community. For example, in the abstract "...to illustrate that membrane potential does not always reflect an increase in mitochondrial oxidative function." Such phrases should be removed (this would help with clarity and readability) especially given they go on to discuss how membrane potential reflects changes in OXPHOS in section 1.
- 5) The authors include JC-1 in the 'non-nerstian mito dyes' section but then state that it could technically be considered a Nernstian dye, except that it forms crystals and aggregates. Thus, it is a nerstian dye (albeit with less sensitivity) but low quality. The authors might consider reclassifying it, while listing all the caveats.
- 6) The authors state that Mitotracker Green does not effectively accumulate in depolarized mitochondria, but then suggest using it for mito mass and normalizing TMRE (Table 1). This should be changed (or at least specify case-use, i.e. polarized mito)
- 7) 2.3. The authors should list 1 or 2 examples of cell lines in which MDRs have been shown to expel mitochondrial dyes.
- 8) Table 1: the authors should include cells for which the dyes do or do not work at the given concentrations.
- 9) Figures. The figures are low quality. The color schematic is unclear. The arrow widths are confusing and not intuitive. Having clearer figures (i.e. clearly labeled mito, color scheme etc) would go a long way in making ETC/MMP relationship accessible.

Referee #1:

This is a detailed methods commentary about the pitfalls of assays for monitoring mitochondrial membrane potential, which will be very useful for any researchers interested in integrating these assays into their work. The authors comprehensively provide guidance for choosing and establishing appropriate assays, which should help improve rigor and interpretability in mitochondrial research. I only have a few suggestions the authors might want to consider for further improving the clarity of the manuscript.

1. For the broad readership of EMBO Journal, it is important to make sure that all relevant terms and concepts are explained. E.g. it would be helpful to define 'Nernstian behavior' and 'uncoupled mitochondria' when these terms are first introduced.

We thank the reviewer for this constructive suggestion. We have now defined the terms **coupled mitochondria**, **uncoupled mitochondria**, **inefficient/leaky mitochondria** and **Nernstian behavior** when they are first mentioned in the text, as well as adding a Glossary section with the most relevant terms.

2. For assessment of mitochondrial mass, would it be helpful to include a brief discussion of mitochondrially targeted fluorescent proteins as organelle markers?

We have now added a new Section 2.3, summarizing the use of mitochondrially targeted fluorescent proteins to control for mitochondrial mass, as well as one example of fluorescent proteins used to estimate membrane potential.

3. Figure 1 is critical for illustrating the commentary's concepts, but this figure is rather difficult to comprehend. I assume the arrows indicate charge flows? It could be helpful to code positive/negative charges with different colors. The text and arrows above the panels seem redundant with the charge flow depicted in the panels - consider to remove text and improve clarity of these processes in the figure panels.

We thank the reviewer for this comment. We have removed text in the Figure to improve clarity. The figure we submitted was a sketch, not a figure in a final form, as the final figure will be generated by a professional illustrator hired by EMBO journal. This text aimed to clarify the message to the illustrator.

4. It might help to split Figure 1 into two figures: 1) Different physiological contexts. Here I find the cell types rather specialized info, consider emphasizing physiological state rather than cell type. 2) Experimental manipulations: This of course includes oligomycin, but it would be very helpful to illustrate other manipulations that are discussed in the text and how they inform on mitochondrial state, e.g. drugs

changing potential, proton gradient, ATP synthesis, ATP/ADP antiport.

This is a very helpful and constructive suggestion. We have split Figure 1 into two new Figures: Figure 1 and Figure 2. Figure 1 shows physiological conditions and Figure 2 shows experimental conditions, with Figure 2 including the divergent effects of oligomycin on the polarization of ATP synthesizing and ATP hydrolyzing mitochondria.

Referee #2:

Mitochondrial membrane potential is frequently used as the sole measure of mitochondrial metabolism and apoptosis, and the measurements are often inadequately performed or interpreted. Therefore, an authoritative review on this topic is much needed, strongly justifying the efforts of Tovar-Ferrero et al. The piece aims to provide 1) some bioenergetics basics to help guide interpretations including why measuring $\Delta\psi$ alone might not be enough to understand bioenergetics status, and 2) important details about commonly used tools to measure $\Delta\psi$, with the intention of minimizing misinterpretations.

In my opinion, the ms needs improvements before it can be considered for publication.

Wild card refers to "uncertainty" whereas the issue often seems to be more one of difficulties in interpretation. Maybe the title/some subtitles would need reconsidering.

We thank the reviewer for this constructive comment. One of the accepted meanings of wild card is: "*someone or something whose behavior is sometimes **unexpected***" (<https://dictionary.cambridge.org/dictionary/english/wild-card>). Thus, we introduced the wild card term not to reflect uncertainty, but to reflect the unexpected hallmarks of mitochondrial membrane potential measurements for researchers not familiar with mitochondrial bioenergetics. This comment however made us realize that wild card is commonly associated to uncertainty as well. Consequently, we agree that this wording will not be clear to readers associating wild cards only with uncertainty. Thus, to ease the interpretation for a larger number of readers, we changed the title to: "***Mitochondrial $\Delta\psi$ hallmarks that enable the accurate assessment of their function in intact cells.***" And the subtitles are now referred to as four hallmarks of $\Delta\psi$, instead of wild cards.

There needs to be a clear statement near the start of the section 1. that $\Delta\psi$ reflects a balance of polarization (H^+ from matrix to IMS side of IMM) and depolarization (H^+ from IMS to matrix side of IMM), and that multiple processes contribute to each (mainly the respiratory complexes pump H^+ from matrix to IMS) and several process move H^+ from IMS to matrix (e.g., ATP synthase, PiC, MPC).

We thank the reviewer for this insightful point. We now open the paragraph of Section 1 with a clearer statement (underlined below) following the reviewer's recommendation: "OXPPOS refers to the process by which mitochondria consume oxygen to synthesize ATP. OXPPOS is constituted by two autonomous modules: the electron transport chain (ETC), which reduces oxygen to water and generates the $\Delta\Psi_m$, and the ATP synthase, which consumes the $\Delta\Psi_m$ (Figure 1A). Electron transport generates the $\Delta\Psi_m$ by driving protons out of the mitochondrial matrix to the intermembrane space (IMS), while ATP synthase consumes the $\Delta\Psi_m$ by bringing protons from the IMS back to the matrix. Importantly, there other mitochondrial transport processes and reactions that affect the proton gradient and the $\Delta\Psi_m$, meaning that these other processes different that ATP synthase and the ETC can impact OXPPOS. Mitochondria in which the $\Delta\Psi_m$ is mostly consumed by the ATP synthase are named coupled mitochondria, as the $\Delta\Psi_m$ generation by the ETC is coupled to $\Delta\Psi_m$ consumption by the ATP synthase."

In the section about the dynamic range of $\Delta\Psi_m$, it should be stated that the actual dynamic range in mitos in intact cells is not really known.

We thank the reviewer for this comment. We have added this statement in Section 1.2.: "Despite the exact $\Delta\Psi_m$ range in mitochondrial from intact cells remains unknown, measurements of the $\Delta\Psi_m$ in intact neurons revealed a range between -108 mV and -156 mV under different physiological challenges."

The Authors try to make the point that the $\Delta\Psi_m$ will generally change only within a narrow range; they could add the caveat that $\Delta\Psi_m$ would dissipate with PTP opening, sustained or flickering.

We agree with the reviewer that flickering and PTP should be added to the review. We were citing this narrow range in Section 1.2, as it is the section focused on coupled mitochondria. Section 1.3 was the one focusing on the physiological conditions that cause $\Delta\Psi_m$ dissipation and uncoupling, such as UCP1 activation. Thus, we included the suggested statements and a key reference in Section 1.3 (underlined), reading as:

"Another important example is the reversible or sustained opening of the mitochondrial permeability transition pore (PTP) in the inner membrane, which can occur in mitochondria from many different cell types to reversibly and irreversibly dissipate the $\Delta\Psi_m$ (Hüser and Blatter, 1999). The irreversible opening of the PTP causes cytochrome c release to kill cells by apoptosis. Thus, observing a large change in $\Delta\Psi_m$ from the resting state will reveal the existence of uncoupling or a complete abrogation of ETC, a scenario incompatible with mitochondrial ATP synthesis activity by OXPPOS."

Most of the classic and many of the current $\Delta\Psi_m$ measurements are carried out in isolated mitochondria or permeabilized cell suspensions in fluorometer cuvettes. I

think this approach has to be described and discussed.

We agree with this reviewer that these approaches are very important, which provided and will provide new and highly relevant knowledge on mitochondrial physiology. Due to space limitations though, and our focus on less sophisticated approaches that we believe can be more interesting for entry-level researchers, we focused our piece exclusively on measurements in intact cells. The reason is that mitochondrial isolation and permeabilization experiments are more complex and might not be preferred by entry-level researchers. To acknowledge this limitation of our manuscript, we included *intact cells* in the new title and we included this statement in the discussion: “*We also want to acknowledge that isolated mitochondria and permeabilized cells can be used to monitor $\Delta\Psi_m$, and we refer readers to these publications (Scaduto RC Jr and Grotyohann, 1999; Fisher-Wellman KH, et al., 2018).*”

Based on the experience of this reviewer and other scientists, MitoTracker green fluorescence is not completely insensitive to $\Delta\Psi_m$ upon accumulation into the mitochondria. This is another limitation in its use as $\Delta\Psi_m$ -insensitive reference.

We appreciate this highly important point raised by the reviewer. This point was only implicit in the previous version of the manuscript, but not explicitly stated. We have now added a new Figure 3 with data from our laboratory showing that laser-induced toxicity can decrease Mitotracker Green staining over time in live cells and that conditions that stress mitochondria can even induce a faster decline. New text has been added as well to explicitly acknowledge this limitation: “*Such Δp dependency combined with binding means that if an excess of MTG molecules that did not bind to mitochondrial proteins is not properly washed, or if changes in the mitochondrial redox occur that break the binding of MTG to mitochondrial proteins, such events can result in a decrease in MTG signal in mitochondria over time, particularly after laser exposure. In Figure 3, we show an illustrative example in which MTG signal is lost with cumulative laser exposure and that this decrease is accelerated in cells with a mitochondrial stress. These data illustrate the need of properly controlling whether MTG labeling is stable for its use as a normalization dye.*”

The ratio fluorescence approach for TMRE/TMRM/R123 has to be described in the review (PMID: 9876159; PMID: 30257218).

This is a great suggestion. We now described the ratiometric approach in Section 2.1.1, as well as in the Conclusion section, highlighting that such an approach is not recommended in intact cells and that additional relevant approaches exist in isolated mitochondria. The focus of this piece being on intact cells was the reason why we initially excluded the citation of this approach in the first version. The new text reads as: “*While a ratiometric approach has been used in isolated mitochondria that improves optimize the specificity of TMRE/TMRM reporting on $\Delta\Psi_m$, this technique is problematic in intact cells and tissues like the perfused heart, where similar spectral shifts occur in the*

cytosol that interfere with the ratio (Scaduto RC Jr and Grotyohann, 1999; Fisher-Wellman KH et al., 2018)."

Also, mentioning that TMRE isn't exactly nernstian and why is needed.

We thank the reviewer for this point. This point was only implicit in the previous version by recommending the use of OXPHOS inhibitors to track non-Nernstian behavior but now we explicit state this in this new text: *"Thus, most of their fluorescence detected in cells with polarized mitochondria stems from mitochondria and has a Nernstian behavior. However, as with other dyes, TMRE/TMRM concentration needs to be optimized, and ideally employed in the lower concentration range. The concentrations and incubation time of TMRE/TMRM to diffuse across membrane and equilibrate without laser induced toxicity and non-Nernstian behavior differ substantially across laboratories (e.g. 15nM for 1h in Wolf et al., 2019 vs 500nM for 10min in Kleele et al., 2021 for TMRE in HeLa cells). For these reasons, we strongly suggest to optimize concentrations and time of dye incubation before image acquisition. More specifically, TMRE/TMRM bound to mitochondrial lipids show a 100-fold increase in fluorescence when compared to free TMRE, meaning that membrane potential insensitive TMRE molecules stuck to lipids can have a great contribution to the fluorescence measured. In addition, the use of high TMRE/TMRM concentrations amplifies laser-induced toxicity and can cause the dye to be in the quenching mode."*

This reviewer would swap sections 3 and 4

Thank you for this great suggestion. We swapped the order and now the quenching section is the third and flow cytometry section is the fourth.

Minor

-Text refers to Fig1 A, B panels but no panel labeling is shown on Fig1.

Furthermore, Figure 1 top panels are not illustrative of what's trying to be explained.

We have revised Figure 1 extensively, including new adequately labeled panels and improving the top panels to provide the intended message in a clearer manner. First, we divided Figure 1 into two Figures. Then we included dials with the different relative values of proton extrusion, proton entry, electron transport and ATP/ADP exchange rates under the different physiological conditions. We think that the use of dials facilitate comparisons between conditions.

For left panel, should show a much fatter arrow for H⁺ matrix->IMS, then, in right panel, the arrow showing H⁺ IMS -> matrix should fatten

We eliminated arrow thickness as a measure of rates, and we used dials instead to illustrate the differences in proton translocation rates under the different physiological conditions. These dials show now larger rates of proton extrusion in left panel and larger rates of proton import in top right panel.

Adding a "dial" scheme to show "net" delta psi (stead state delta psi) would be helpful: highly polarized for substrate only, then slightly depol for a little ADP, then more depol with more ADP

We have now included the mV values under each condition determined in published papers, to illustrate the slight depolarization with more ADP in Figure 1, as well as to illustrate the differences in ATP hydrolyzing and ATP synthesizing mitochondria under oligomycin in new Figure 2

-The means of mitochondrial membrane Pg1 last para
The transfer of electrons across complexes I, III and IV requires proton translocation across the inner membrane. To correct content, please replace requires with "drives"

"requires" has been replaced by "drives": *"The transfer of electrons across complexes I, III and IV drives proton translocation across the inner membrane."*

-Pg3 para1
higher protonophore concentrations that completely dissipate $\Delta\Psi_m$ decrease respiration
delete "that"

The sentence reads now: *"higher protonophore concentrations completely dissipate $\Delta\Psi_m$ and decrease respiration"*

-pg3 para 3
In an oxygen consumption experiment, this could be seen as increased basal and proton-leak driven oxygen consumption rates, without observing differences in absolute oligomycin-sensitive respiration rates
Not very clear

We rewrote this sentence, trying to make it clearer. Now it reads as: *"Such a scenario would be revealed by an increase in basal respiration associated with a decrease in $\Delta\Psi_m$, in the absence of changes in ATP-producing respiration."*

-pg4 para2
ATP synthase might be activated in excess in some pathologies, inducing ATP wasting
is it really meant that the pathology is "ATP wasting"?

We re-wrote the sentence to be less equivocal and reflect what the original publication concluded: *"In this regard, we recently identified that, in some pathologies, the excessive activation of the reverse mode of the ATP synthase that hydrolyses ATP is maladaptive"*

Referee #3:

The central role of mitochondria in the signaling and metabolic processes of a eukaryotic cell have led to an increasing interest in mitochondrial bioenergetics. At the core of mitochondrial bioenergetics is mitochondrial membrane potential (MMP), the voltage differences across the inner mitochondrial membrane. The manuscript by Tovar-Ferrero et al provides a manual that will guide mitochondrial researchers in the appropriate tools and interpretation of commercially available fluorescent dyes that enable MMP measurements. The manuscript is timely and clear, and will be of great use to the community. The authors might consider some changes for clarity and utility.

Minor:

1) Organization. The authors might consider restructuring their manuscript with a first section 1 dedicated to first explaining mitochondrial OXPHOS basics. This might help all readers know the fundamentals, thereby enabling them to better understand the details of the relationship between OXPHOS and MMP. Then, a section 2 dedicated to understanding how to use and interpret commercially available dyes.

We appreciate this point raised by the reviewer. To make it easier to understand for a wider audience, we modified the first paragraph of Section 1.1 to explain the key basics of OXPHOS. In this regard, Section 1 is exclusively dedicated to OXPHOS, which is divided in 3 subsections that cite key physiological conditions with divergent changes in membrane potential and OXPHOS. Section 2 is focused on explaining how the membrane potential dyes work, which are the different types available, how to use them and how to know when their fluorescence report on MMP.

2) It is unclear why/what the authors are referring to with 'wild card.' A wild card refers to a person or thing whose influence is unpredictable and whose qualities are uncertain. As the authors discuss, the dyes are not so much unpredictable as incorrectly used. The authors might consider a simpler title, for example 'A guide to using membrane potential assays.' The continuous references to 'wild cards' are confusing and unclear. The authors might use 'considerations' rather than wild-cards.

We thank the reviewer for this insightful suggestion, shared with Reviewer 1. As mentioned in the answer to Reviewer 1, these four wild cards are the hallmarks that, in our experience, were unexpected for most researchers asking for advice on these assays, as well as being the most commonly missed hallmarks in papers we received for review. Given that both Reviewer 1 and this Reviewer thought of uncertainty instead of something unexpected, we changed the title to "***Mitochondrial $\Delta\Psi_m$ hallmarks that enable the accurate assessment of their function in intact cells.***"

3) The authors should more clearly explain proton extrusion and proton import.

We appreciate this point raised by the reviewer, and rephrased the first paragraph in Section 1.1 to clearly explain proton extrusion and import. This paragraph now reads as:

OXPHOS refers to the process by which mitochondria consume oxygen to synthesize ATP. OXPHOS is constituted by two autonomous modules: the electron transport chain (ETC), which reduces oxygen to water and generates the $\Delta\Psi_m$, and the ATP synthase, which consumes the $\Delta\Psi_m$ (Figure 1A). Electron transport generates the $\Delta\Psi_m$ by driving protons out of the mitochondrial matrix to the intermembrane space (IMS), while ATP synthase consumes the $\Delta\Psi_m$ by bringing protons from the IMS back to the matrix. Importantly, there other mitochondrial transport processes and reactions that affect the proton gradient and the $\Delta\Psi_m$, meaning that these other processes different that ATP synthase and the ETC can impact OXPHOS. Mitochondria in which the $\Delta\Psi_m$ is mostly consumed by the ATP synthase are named coupled mitochondria, as the $\Delta\Psi_m$ generation by the ETC is coupled to $\Delta\Psi_m$ consumption by the ATP synthase.

4) The authors often make statements that make assumptions regarding prevailing beliefs in the community. For example, in the abstract "...to illustrate that membrane potential does not always reflect an increase in mitochondrial oxidative function." Such phrases should be removed (this would help with clarity and readability) especially given they go on to discuss how membrane potential reflects changes in OXPHOS in Section 1.

We now removed this sentence from the paragraph, as well as rewriting other sentences that might be unclear as well. Please see the previous answer with the new opening paragraph in Section 1, as well as new sentences that we re-wrote across the manuscript marked in the track changes document.

5) The authors include JC-1 in the 'non-nerstian mito dyes' section but then state that it could technically be considered a Nernstian dye, except that it forms crystals and aggregates. Thus, it is a nerstian dye (albeit with less sensitivity) but low quality. The authors might consider reclassifying it, while listing all the caveats.

We agree with the reviewer. Accordingly, we defined JC-1 as semi-nerstian in the previous version of the manuscript. Thus, we reclassified it as a Nernstian dye with this revised sentence: "*As a result, we can conclude that JC-1 is a dye with lower sensitivity, specificity and reliability to measure changes in Ψ_m , when compared to other dyes. Thus, JC-1 could be even considered a dye mostly behaving in a non-Nernstian manner.*"

6) The authors state that Mitotracker Green does not effectively accumulate in depolarized mitochondria, but then suggest using it for mito mass and normalizing

TMRE (Table 1). This should be changed (or at least specify case-use, i.e. polarized mito)

We now specified in the table that MitoTracker Green is effective only in cells that retain some levels of polarization in their mitochondria. In addition, we added a novel Figure showing that MitoTracker Green is not infallible and its staining can be lost faster after laser exposure in stressed mitochondria (New Fig. 3), as well as when unbound molecules are not properly washed.

7) 2.3. The authors should list 1 or 2 examples of cell lines in which MDRs have been shown to expel mitochondrial dyes.

We now added the 2 specific examples, Hep-2 and hematopoietic stem cells, as cells expelling mitochondrial dyes by MDRs, and added a new specific reference for HSC cells (Almedia et al, 2017).

8) Table 1: the authors should include cells for which the dyes do or do not work at the given concentrations.

We agree with the reviewer's suggestion that including the appropriate working concentrations for each dye, and particularly, in which cells these dyes do work would be useful. We added an extra section in Table 1 indicating some of the working concentration described for some cell lines that work. However, we would like to emphasize that there are multiple variables involved in determining whether these dyes report on membrane potential with specificity and sensitivity. For example, we incubate TMRE at 15nM or 200nM for 1h for live imaging of HeLa or MEFs cells (Wolf et al., 2021, Acin-Perez et al., 2023), whereas Kleele et al., 2021 incubates the same dye at 500nM for 10min, also in HeLa cells. This means that each laboratory should try different concentrations first, with and without MDR inhibitors, and determine whether they can successfully detect fluorescence that is sensitive to MMP. Thus, providing a list of what works in our hands might deter users of performing these validation experiments first, which are essential.

9) Figures. The figures are low quality. The color schematic is unclear. The arrow widths are confusing and not intuitive. Having clearer figures (i.e. clearly labeled mito, color scheme etc) would go a long way in making ETC/MMP relationship accessible.

We thank the reviewer for this comment. Part of this lack of clarity is that these figures are just a sketch. These Figures were not submitted in a final format, as they will be re-done by a professional illustrator working for the EMBO journal. We have now changed the arrows by dials to illustrate the differences in flux better, as well as splitting Figure 1 in two Figures as suggested by Reviewer 1. We believe that the

message is clearer in this current format and will enable to illustrator to generate a better Figure.

Dear Marc,

Thank you for your patience with the experts' feedback. We have now received additional comments by two of the colleagues. As you will see they state that the commentary has been substantially improved by your amendments, and they recommend publication, pending minor remaining changes.

Hence, I am happy to share that your Comment is provisionally accepted for publication at the EMBO Journal.

Please revisit the additional remarks by expert #2 and amend the text where appropriate. I will myself also go again through the manuscript text and suggest final minor changes and edits shortly.

Also, I kindly ask you to consider formatting requests as to the list enclosed below.

There is an additional matter to be solved about Figure 3 and the data contained. We need you to either deposit this data to a freely accessible database such as Dryad or Figshare or similar, and annotate the methods used and statistics accordingly; alternatively you might either refer to an example of already published data or omit the figure and related reference, rather leaving the general statement in the text.

Finally, I will now share the other Figure drafts with our scientific graphics illustrator, who will approach you shortly on figure versions translated into journal style.

I look forward to your last revision and moving ahead with formal acceptance and publication of this commentary shortly.

Best wishes,

Daniel

Daniel Klimmeck, PhD
Senior Editor | The EMBO Journal
d.klimmeck@embojournal.org

>> please check callouts: Fig 2A is called out before Fig 1B.

>> Funding: please enter 'ERDF A way of making Europe" by the European Union' into our online system.

Referee #1:

The authors revised the figures and manuscript for clarity and completeness. This is a thoughtful and informative review that will be relevant to any researchers interested in measuring the mitochondrial membrane potential in mammalian cells.

Referee #2:

The Authors made amendments following the referee suggestions and the ms has improved. The text still needs to be carefully checked both for clarity and grammar. These issues became more apparent after the Authors' revision. As an example a few problems are listed below for section 1.1. Please address these issues and also check and amend the rest of the text:
1.1.

Importantly, there ARE other mitochondrial transport processes and reactions that affect the proton gradient and the $\Delta\Psi_m$, meaning that these other processes different that ATP synthase and the ETC can impact OXPHOS.
Sentence needs revision. Missing verb has to be added

But how DOES the consumption of $\Delta\Psi_m$ by ATP synthase controls ETC and therefore mitochondrial oxygen consumption?
Missing verb in this sentence, too.

If proton translocation does not occur, oxygen consumption will be impeded
O₂ reduction is bc of transfer of electrons, not pumping of protons. Pls clarify what you mean/add reference. Sentence seems to be dispensable anyways.

The fact that electron transfer drives the translocation of protons from the matrix to the intermembrane space explains why higher oxygen consumption can sometimes result in an increase of the p and Ψ_m .
The sentence is not ordered logically. Please include reference.

2 paragraphs later:

-oxygen consumption can result in lower Ψ_m

- Thus, higher oxygen consumption can result in increased Ψ_m if the elevation in electron transfer rates is larger than the capacity of ATP synthase to produce ATP

These statements together can create confusion about the relationship between JO₂ and $\Delta\Psi_m$. At least "can result in" would need to be replaced with "can be associated with" but more clarity on the cause effect relationships between electron transfer, JO₂, proton pumping and $\Delta\Psi_m$ would benefit the readers.

The authors addressed the minor formatting issues.

Dear Marc,

Thank you for sending us the updated final version of the commentary article.

I am pleased to inform you that your manuscript has been accepted for publication in the EMBO Journal.

Your manuscript will be processed for publication by EMBO Press. It will be copy edited and you will receive page proofs prior to publication. Please note that you will be contacted by Springer Nature Author Services to complete licensing and payment information. Please note that as this is invited front-half content, OA charges applicable to this article will be covered. Please use the following token - XXXXXXXXXXXXX- when entering the licensing process.

Further, as mentioned our graphics illustrator team is currently converting the commentary figures into journal style. They will contact you shortly on the final proof stage figures versions for your input.

If you have any questions, please do not hesitate to contact me.

Thank you again for your kind contribution to The EMBO Journal, which is much appreciated.

Best regards,

Daniel

Daniel Klimmeck, PhD
Senior Editor
The EMBO Journal